# Sensitivity of Viscoelastic Tests to Platelet Function

**DOI:** 10.3390/jcm9010189

**Published:** 2020-01-10

**Authors:** Marco Ranucci, Ekaterina Baryshnikova

**Affiliations:** Department of Cardiothoracic, Vascular Anesthesia and Intensive Care, IRCCS Policlinico San Donato, 20097 Milan, Italy; ekaterina.baryshnikova@gmail.com

**Keywords:** platelets, platelet function tests, viscoelastic methods

## Abstract

Viscoelastic tests provide a dynamic assessment of coagulation, by exploring the time to clot formation and the clot strength. Using specific activators or inhibitors, additional factors can be explored, like the fibrinogen contribution to clot strength. Since the early days, various attempts have been done to measure platelet function with viscoelastic test. In general, the difference between the maximum clot strength and the fibrinogen contribution is considered an index of platelet contribution. However, this parameter does not clearly split platelet count from function; additionally, the extensive thrombin generation of standard activated viscoelastic tests activates platelet through the protease activated receptors, bypassing the other pathways. For this reason, standard viscoelastic tests cannot be used to assess platelet reactivity under the effects of aspirin or P2Y_12_ inhibitors. To overcome this limitation, a specific test was developed (thromboelastography platelet mapping). This test has been compared with the gold standard of light transmission aggregometry and with other point-of-care tests, with conflicting results. In general, the use of viscoelastic tests to assess the effects of antiplatelet agents is still limited. Conversely, platelet contribution to clot strength in the setting of coagulopathic bleeding is considered an important parameter to trigger platelet transfusion or desmopressin.

## 1. The Role of Platelets in the Viscoelastic Pattern of Blood Coagulation

Since the very beginning of thromboelastography (TEG), the classical tracing of viscoelastic tests (VET) is that represented in Figure 1. Basically, it reflects the original tracing produced with a mechanical device by Hartert and associates in 1948 [1], and subsequently replicated with a different methodology in thromboelastometry (TEM) [2]. In the first-generation devices, the basic principle is the transmission of movement from rotating blood to a torsion-wired suspended pin, or the friction of blood on a rotating pin. In both cases, until the blood remains in the fluid phase, no movement is transmitted to the suspended pin nor the rotating pin is braked in its movement. This corresponds to the flat line that is the first part of both the TEG and the TEM tracings, whose normal duration varies from 1 to 8 min depending on the activator used. During this period, the physical properties on blood do not change, while the intrinsic and/or extrinsic coagulation cascades concur in thrombin generation. The blood starts interacting with the pin once the first fibrin fibers are formed, with subsequent platelet aggregation and final formation of a stable clot (stabilized by activated factor XIII). The physical status of the clot is that of a gel, and the transition point from the liquid to the gel phase may be called the “gel point”. The more firm and stiff the clot is, the more it will transmit or brake the movement to the pin or of the pin, and this translates, in graphical form, in a larger maximum amplitude (MA, in TEG) or maximum clot firmness (MCF in TEM). Platelets are the main contributor to MA/MCF, and therefore both platelet count and function should be reflected by these parameters. However, the assessment of platelet reactivity (function) is not simply expressed by the MA/MCF for a number of reasons that will be addressed in this Review.

## 2. Historical Background of Platelet Function Analysis with VET

Since the very beginning, different authors investigated the effects of different platelet count and function on the VET tracings. Early in 1962, Parreira [3] observed that the MA at TEG was directly proportional to the platelet count. This observation was confirmed by subsequent studies, where the clot firmness demonstrated a direct logarithmic relationship with platelet count [4]. In 1971, Hawkins [5] tested with TEG 17 subjects under aspirin compared to normal controls. She found that the reaction time (R) and the K time were prolonged (almost doubled) in aspirin treated patients, while no difference in MA was found. These results were subsequently contradicted in a study on pregnant and non-pregnant subjects after aspirin ingestion, where the authors could not observe any change in TEG parameters after ingestion of 600 mg of aspirin [6]. Same results were observed by Trentalange and associates [7] in a study on healthy volunteers receiving 650 mg of aspirin.

The effects of antiplatelet agents acting on the P2Y_12_ receptor (clopidogrel, prasugrel, ticagrelor) have been studied with VET. In an elegant study, Tanaka and associates demonstrated that despite an in-vitro complete inhibition of ADP-dependent platelet aggregation, TEG tracings remained unchanged, and that the same happened in clopidogrel-treated patients [8]. Gurbel and associates found a significant R-time prolongation after clopidogrel loading dose [9]; however, the values remained within the normal range. Overall, the existing evidence is that standard TEG and ROTEM are insensitive to a drug-induced inhibition of platelet aggregation through the thromboxane pathway or the P2Y_12_ receptor. Measures of clot firmness like the MA and the MCF are decreased only in the case of GPIIbIIIa receptor inhibition, reflecting a poor interaction of platelets into the fibrin network [10].

## 3. Why Standard VET Are Inadequate to Test Platelet Reactivity under Anti-Platelet Agents

VET-based assessment of the dynamic process of coagulation includes the use of different types of activators (kaolin, ellagic acid, tissue factor, and others) at high doses, to trigger and accelerate the coagulation process. This invariably results in a burst of thrombin generation. Platelets represent a complex environment, with a number of different possible pathways leading to activation and aggregation. Basically, aggregation is promoted by the final receptor GPIIbIIIa, which links fibrinogen and promotes platelet-to-platelet aggregation. Drugs acting on this receptor (GPIIbIIIa inhibitors) promote a powerful platelet inhibition, and VET are sensitive to this action, showing decreased levels of clot firmness. Things are different for drugs acting on the thromboxane pathway (aspirin) or the P2Y_12_ ADP-dependent receptor (thienopyridines and ticagrelor): the thrombin generated by the activators strongly interacts with the protease-activated receptors (PAR) that are powerful triggers for platelet activation. Activation of PAR masks the pharmacological inhibition of ADP-dependent receptors, bypassing them. For this reason, the standard VET are insensitive to the pharmacological effects of aspirin, thienopyridines, and ticagrelor. In conclusion, as already pointed out by other authors, standard VET do not provide a comprehensive or sensitive reflection of pharmacologically impaired platelet function [10].

## 4. VET, Platelet Function, and Direct Oral Anticoagulants (DOAC)

DOAC (rivaroxaban, apixaban, edoxaban, and dabigatran) limit thrombin generation by inhibiting factor Xa or factor IIa (thrombin). Thrombin is a powerful activator of PAR, and hence thrombin inhibition results in a decreased platelet reactivity. Studies using thrombin generation test (TGT) showed that in platelet-rich plasma with prior collagen-induced platelet aggregation TGT includes platelet ability to support thrombin generation in the presence of direct factor Xa inhibitors (rivaroxaban) [11]. Many studies addressed VET ability to detect the effects of DOAC, generally showing a prolongation of the clot formation time (R-time or CT) [12]. VET-based parameters incorporating platelet contribution are the MA and the MCF, and conflicting results exist with respect to MA and MCF changes under DOAC and after DOAC reversal. Xu and associates [13] reported minimal changes of MA under dabigatran effect, while Escolar and associates found that rivaroxaban decreased the MCF due to a poor platelet-fibrin interaction [14]. However, no study addressed the separate contribution of fibrin (ogen) and platelets to the clot firmness under DOAC effects, and the use of VET in the setting of DOAC treatment remains limited.

## 5. Splitting Platelet and Fibrin Contribution: The Platelet Component

The maximum clot firmness (or stiffness) is expressed by the MA at TEG and the MCF in ROTEM in terms of amplitude; new devices (Quantra, HemoSonics LLC, Charlottesville, VA, USA) provide the same measure (clot stiffness, CS) in terms of shear modulus (hPa). Clot stiffness is a function of the interaction between platelets and fibrin (ogen), stabilized by FXIIIa. This leads to the concept that by subtracting the fibrinogen contribution from the MCF or MA or CS the remaining measure is related to the platelet contribution (PCS), as shown in Figure 2. This concept has been recently incorporated in the Society of Cardiovascular Anesthesiologists Guidelines on the management of perioperative bleeding and hemostasis in cardiac surgery patients [15]. The suggestion is to consider platelet concentrate transfusion and/or desmopressin when the MA or the MCF are low (<40 mm) and the fibrinogen contribution is normal (>8 and >10 mm, respectively). This corresponds to a PCS < 75%.

However, it must be considered that this approach, even if apparently logical, suffers from a certain degree of bias. Shear modulus (or elasticity) is not linearly associated with clot amplitude, being rather defined by the equation [16]:

Shear modulus (G, [dyn/cm²] = (5000 × amplitude [mm])/(100 − amplitude [mm])), that is graphically shown in Figure 3.

Therefore, the PCS simply measured as the difference in amplitude may (even greatly) differ from the true difference in shear modulus, as shown in Table 1. At any given value, PCS based on the difference in amplitude underestimates the true PCS based on the difference in the modulus; given the exponential shape of the relationship, the larger is clot amplitude, the higher is the difference between the two measures. This, in part, limits the clinical impact of this bias, because PCS has a clinical relevance in bleeding patients with a low clot amplitude. PCS is provided however directly as a difference in modulus in the Quantra system, while it needs appropriate calculations in TEG and ROTEM.

Once PCS is correctly measured, it remains representative of the combination of platelet count and function, without the possibility to split one component from the other. Attempts have been done to identify how much of PCS depends on count and how much on function. In a recent study [17], we investigated a series of 103 patients undergoing cardiac surgery. One-hundred-thirty-three measures of ROTEM were available, with simultaneous measure of platelet count and platelet reactivity (multiple electrode aggregometry, MEA) to the stimulation of the ADP (ADPtest) and thrombin activated peptide (TRAPtest) receptors. We could demonstrate that PCS measured as difference in clot shear modulus (PCSel) showed a better association with platelet count and function than PCS measured as difference in amplitude; additionally, we tested the association between PCSel and platelet count and function in a multivariable model. The squared correlation coefficient for association between PCSel and platelet count was 0.36 (*p* = 0.001) and 0.14 (*p* = 0.001) for the association between PCSel and platelet reactivity at the ADPtest. Overall, this means that 36% of the variance of PCSel is due to platelet count, and 14% to platelet reactivity, with the remaining 50% that is unexplained by platelet count or reactivity. Based on these results, the conclusion is that even applying the best methodology to assess PCS, this measure remains poorly associated with platelet function.

In a recent in-vitro study [18], the authors tested blood from 8 healthy volunteers with the TEG 6S device. They investigated the relationship between clot amplitude (MA) and platelet count. Additionally, they induced different degrees of platelet inhibition using abciximab, and measured the relationship between platelet inhibition (%) and MA. They could observe that MA linearly changes with changes in platelet count within values comprised between 28,000 and 91,000 cell/µL. Outside this range, there were no platelet count-dependent MA changes. With respect to platelet function, the changes in MA linearly reflected the changes in platelet inhibition within the range of 68% to 82% inhibition, being insensitive to lower or higher degrees of platelet inhibition.

Recently, parameters of TEG have been tested for association with validated measures of platelet function in healthy volunteers and patients free from antiplatelet agents [19]. Platelet function was measured with whole-blood platelet lumiaggregation and platelet-rich plasma aggregation. The MA at standard TEG was directly correlated with whole-blood lumiaggregation after stimulation with collagen, ristocetin, arachidonic acid, and ADP.

These results are certainly reflecting the fact that the inhibition of the terminal receptor of platelet aggregation GPIIbIIIa has an effect on VET-based clot stiffness, and that in absence of anti-platelet agents the MA/MCF correlates with platelet aggregation. However, platelet dysfunction exerted by other drugs (aspirin, P2Y_12_ inhibitors) or due to other reasons (i.e., the deleterious effects of cardiopulmonary bypass, CPB) remains poorly reflected by changes in PCS.

## 6. Modified VET: TEG Platelet Mapping

The need to provide a measure of platelet aggregation under the effects of P2Y_12_ inhibitors became more and more important with the wide diffusion of double anti-platelet therapy (DAPT). Therefore, the manufacturers of TEG released a modified test called TEG Platelet Mapping (TEG-PM) about 15 years ago [20]. This test is based on a sequence of procedures, that in the TEG 6S are now incorporated in a single cartridge, whereas using the previous TEG version, two devices where needed to run parallel tests. Briefly, a standard kaolin-activated TEG is considered as the “best platelet reactivity test” and the correspondent MA is considered as 100% of platelet function. In a second test, heparin is added to the blood so to blunt the thrombin-dependent clot formation and platelet activation, and a pure fibrin clot is elicited by adding reptilase, which directly converts fibrinogen into fibrin. The MA correspondent to this fibrin clot is considered 0% of platelet function. Subsequently, different platelet activators (arachidonic acid and ADP) are added to the same heparin-based environment. The MA obtained by adding these activators is considered as the platelet aggregation in presence of aspirin or P2Y_12_ inhibitors (Figure 4).

Platelet reactivity (%) is calculated as: 100 × MA (activator)/(MA [kaolin] − MA [fibrin]) and platelet inhibition (%) as 100 − platelet reactivity.

From the theoretical point of view, TEG-PM may offer some advantages over other point-of-care platelet function tests, like MEA, which are strongly dependent on platelet count [21]; conversely, by providing a platelet function measure that is expressed in % rather than in absolute values, TEG-PM is less affected by platelet count.

TEG-PM has been tested against other platelet function analyzers, including the gold standard (light transmission aggregometry, LTA), in subjects with or without the effects of antiplatelet agents.

The effects of CPB on platelet function were studied by Ellis and associates in 2016 [22]; the authors measured platelet function with TEG, TEG-PM, and MEA before and after CPB. All the tests showed a decreased platelet function after CPB; however, TEG and TEG-PM values did not predict bleeding after cardiac surgery.

Trauma patients with trauma-induced coagulopathy were investigated in a recent study, with simultaneous measure of TEG-PM and MEA [23]. The authors found no significant correlation between MEA ADP test and ADP TEG-PM, and a moderate correlation between results obtained by activating the arachidonic acid pathway. In this context, TEG-PM was predictive of blood product transfusion whereas MEA was not.

In a large study on healthy volunteers and donors on daily antiplatelet therapy (aspirin and/or clopidogrel), TEG-PM was compared to LTA, MEA, and VerifyNow [24]. In healthy volunteers, TEG-PM platelet aggregation was 5-times higher than those of donors taking aspirin, with discrimination at receiver operating characteristic (ROC) analysis of 1.000. Conversely, TEG-PM platelet aggregation of healthy volunteers was only slightly higher (85% vs. 72%) than those of donors taking clopidogrel, with a very poor discrimination at ROC analysis (area under the curve 0.589). The authors concluded that TEG-PM is least suited to monitor effects of antiplatelet agents, even due a low intra- and inter-assay precision.

In the setting of patients receiving clopidogrel after carotid artery stenting [25], the authors compared VerifyNow and TEG-PM. Therapeutic goal was considered a value of platelet response units < 194 units at VerifyNow and an MA at the ADP test < 50 mm at TEG-PM. The two tests had a correlation coefficient of 0.50 (*p* = 0.0026); the rate of clopidogrel-resistant patients was 9% at TEG-PM and 39% at VerifyNow. The authors concluded that the agreement between tests was poor, and that TEG-PM detected rate of clopidogrel resistance was more reasonable and in agreement with the existing literature.

Detection of antiplatelet agents in adult trauma patients was explored by Connelly and associates [26]. Out of the 64 patients enrolled, 25 were taking antiplatelet agents (mainly aspirin). TEG-PM with arachidonic acid could identify the use of antiplatelet agents with an area under the curve of 0.90, and had a good correlation with VerifyNow ARU and MEA AspiTest.

Finally, antiplatelet therapy in ventricular assist device patients was investigated with MEA and TEG-PM [27]. There was a poor correlation between the two tests both at the arachidonic acid and the ADP tests. The MEA ADP test showed a good correlation with the clopidogrel dose, whereas TEG-PM did not. The rate of aspirin and clopidogrel resistant patients was much higher in TEG-PM than in MEA tests. The authors concluded that in children with ventricular assist device MEA is more reliable than TEG-PM in monitoring antiplatelet therapy.

## 7. The Clinical Scenarios

There are basically two clinical scenarios for point-of-care measure of platelet function. The first one pertains platelet reactivity in patients under antiplatelet agents, in order to monitor the adequacy of the therapy, to detect drug-resistant patients, to settle the timing for major surgery (namely cardiac surgery [28]). The second scenario pertains the identification of the nature of coagulophatic bleeding after major surgery, trauma, and other clinical entities. Within these scenarios, the role of VET-based platelet function assessment strongly differs.

In the first setting (detection of drug-induced platelet dysfunction) the standard TEG and ROTEM tests are clearly inadequate to assess the effects of antiplatelet agents. The extensive thrombin generation during the tests invariably activates the PAR receptors, masking the effects of aspirin and P2Y_12_ inhibitors by generating normal MA and MCF. Only the inhibition of the final receptor GPIIbIIIa may be detected by standard VET tests.

TEG-PM was designed to overcome this bias; however, its ability in detecting the effects of antiplatelet agents remains controversial, when compared to the more commonly used tests (PFA-100, MEA, VerifyNow). Data in literature report either a good [25] or a poor [27] correlation between different tests, no correlation between drug dose and TEG-PM values [27], a lower [25], or higher [27] rate of drug-resistant patients.

In the second scenario, standard TEG and ROTEM, and namely PCS, seem to reflect bleeding due to a poor platelet contribution to clot strength, and this measure is presently included in some of the existing guidelines [15]. PCS is not able to separate the respective role of platelet count and function, however it seems to reflect more count than function [17]. From a practical perspective, however, addressing the relative role of thrombocytopenia and platelet dysfunction in the bleeding patient is not of paramount importance, since in both cases the only suggested therapy is platelet concentrate transfusion and/or desmopressin [15,28].

## 8. Conclusions

Measuring platelet function is always a tricky task. Platelets are an incredibly complex environment, and different tests explore different abilities, measuring platelet function based on different abilities (time to closure of a hole in a membrane; electrical impedance changes due to aggregation to electric filaments; and, in VET, platelet contribution to clot elasticity). No surprise, therefore, that correlation between different tests is often poor.

To summarize the existing evidence, standard VET are inadequate to assess the effects of antiplatelet drugs like aspirin or P2Y_12_ inhibitors; the confounding effect of DOAC on platelet function is yet to be established, and TEG-PM offers a potential tool overcoming some of these problems. PCS includes both the effects of platelet count and function, and clinical data suggest that measuring PCS is an important part of the algorithms of diagnosis and treatment of the coagulopathic bleeding patient. At present, however, a large and possibly multi-center cohort study linking PCS to bleeding outcomes and finding cut-off values for platelet transfusion in different clinical scenarios is lacking, making highly suggestable to fill this gap in knowledge.

## Figures and Tables

**Figure 1 jcm-09-00189-f001:**
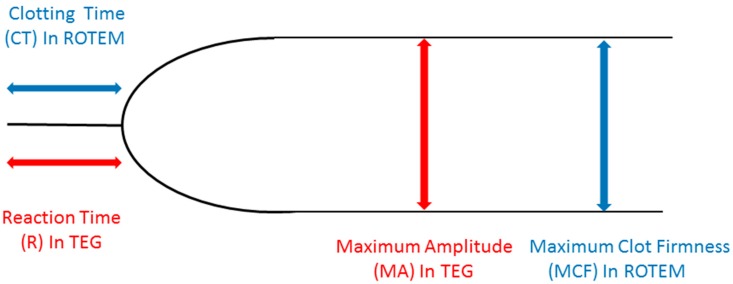
A standard viscoelastic test tracing. ROTEM: rotational thromboelastometry; TEG: thromboelastography.

**Figure 2 jcm-09-00189-f002:**
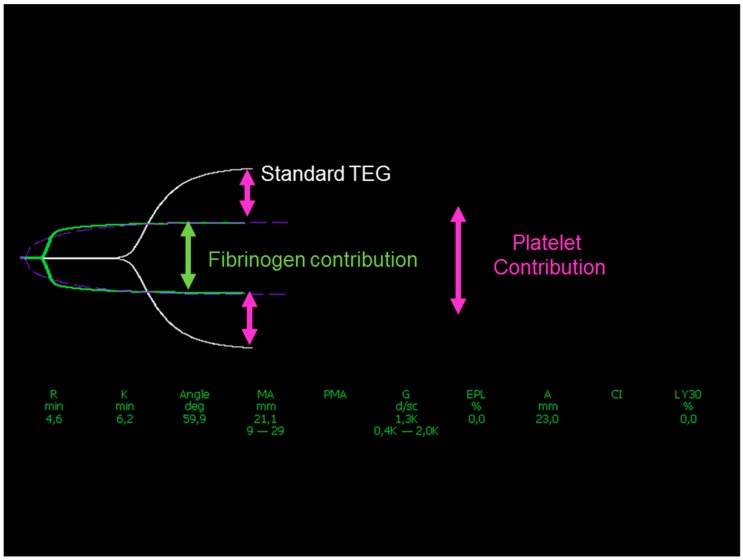
Platelet contribution to clot strength as the difference between amplitudes in standard thromboelastography (TEG, white trace) and fibrinogen contribution (green trace).

**Figure 3 jcm-09-00189-f003:**
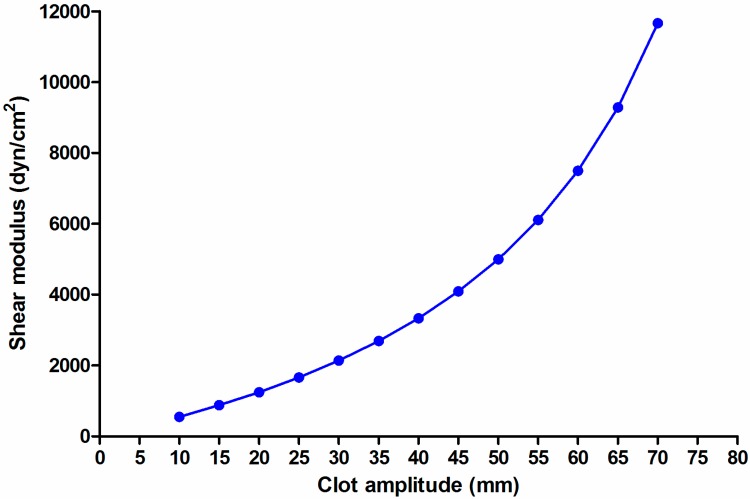
Non-linear relationship between clot amplitude and shear modulus.

**Figure 4 jcm-09-00189-f004:**
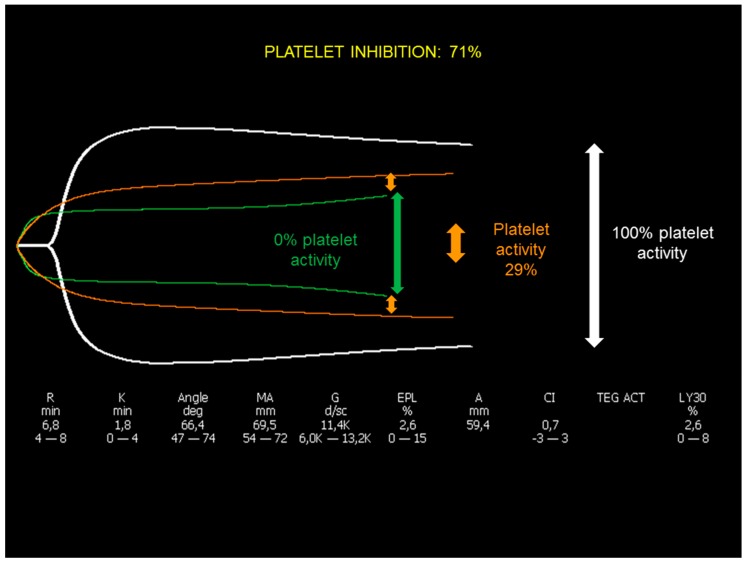
Thromboelastography (TEG) platelet mapping. White trace is standard TEG activated with kaolin; green trace is a platelet-free clot obtained by adding heparin and reptilase; orange tracing is the platelet activity, adding a specific platelet activator (arachidonic acid or ADP).

**Table 1 jcm-09-00189-t001:** Difference in platelet contribution to clot stiffness as measured based on amplitude or elasticity in viscoelastic tests. Calculations based on a fixed (normal) fibrinogen contribution to clot strength settled at 10 mm of amplitude.

Clot Stiffness	Clot Stiffness	FCS	FCS	PCSamp	PCSmod
(amplitude, mm)	(modulus, dyn/cm²)	(amplitude, mm)	(modulus, dyn/cm²)	(%)	(%)
20	1250	10	556	50	55
25	1250	10	556	60	67
30	1250	10	556	67	74
35	2692	10	556	71	79
40	3333	10	556	75	83
45	4091	10	556	78	86
50	5000	10	556	80	89
55	6111	10	556	82	91
60	7500	10	556	83	93
65	9286	10	556	85	94
70	11,667	10	556	86	95

FCS: fibrinogen contribution to clot stiffness; PCSamp: platelet contribution based on difference in amplitude; PCSmod: platelet contribution based on difference in modulus.

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
