# Peer review of "Sensitivity of Viscoelastic Tests to Platelet Function"

_jcm, 2020, doi:10.3390/jcm9010189_

Round 1

Reviewer 1 Report

The review of Ranucci and Baryshnikova summarized the current data about the use of TEG and TEM to assess platelet function. The manuscript is concise and well written and give a good overview of the current knowledge.

In my opinion the review would benefit some data about the current problems encountered when using viscoelastic tests in the presence of new oral anticoagulants (rivaroxaban, apixaban, dabigatran) Indeed FXa and thrombin inhibitors can affect platelet activation (and many patients under oral anticoagulant are also under antiplatelet drug) (Pailleret, Eur J Anaesthsiol 2019; Taune, J Thromb Haemost 2018…). In addition, some more context could be added regarding other methods (e.g. thrombin generation) giving informations on platelet function in the presence of antiplatelets or anticoagulants (Makhoul, J Clin Med, 2019; Tarandovskis, PloS One 2013…).

Minor:

Line 29: In the first-generation devices

Figure 2: do not use Comics Sans MS in your figure

Line 163: function

Line 177: “best platelet reactivity test”

Author Response

The review of Ranucci and Baryshnikova summarized the current data about the use of TEG and TEM to assess platelet function. The manuscript is concise and well written and give a good overview of the current knowledge.

Reply: we thank the Reviewer for the appreciation

In my opinion the review would benefit some data about the current problems encountered when using viscoelastic tests in the presence of new oral anticoagulants (rivaroxaban, apixaban, dabigatran) Indeed FXa and thrombin inhibitors can affect platelet activation (and many patients under oral anticoagulant are also under antiplatelet drug) (Pailleret, Eur J Anaesthsiol 2019; Taune, J Thromb Haemost 2018…).

Reply: we have now added a subchapter dedicated to this item

In addition, some more context could be added regarding other methods (e.g. thrombin generation) giving informations on platelet function in the presence of antiplatelets or anticoagulants (Makhoul, J Clin Med, 2019; Tarandovskis, PloS One 2013…).

Reply: we briefly mentioned TGT in the subchapter; however, not being a viscoelastic test, TGT is outside the topic of the manuscript.

Minor:

Line 29: In the first-generation devices

Reply: done

Figure 2: do not use Comics Sans MS in your figure

Reply: done

Line 163: function

Reply: done

Line 177: “best platelet reactivity test”

Reply: done

Reviewer 2 Report

The manuscript by Ranucci & Baryshnikova aims to provide a review on sensitivity of viscoelastic tests to platelet function.  The work could potentially be interesting overview but at the current stage has major issues that need to be addressed.

English language and grammar need to be revisited. Certain sentences are lacking propositions and are using colloquial words that should be omitted in scientific publication. The objective of this review is to talk about viscoelastic test and platelet function. However platelet function is discussed in great limitations. In fact there is no mention of molecular targets of aspirin in platelets and how that may affect function. Overall a brief discussion of coag and platelets should be introduced Each parameter of the various tests needs to be defined and discussed in terms of platelet function. Summary of findings should be not only stated but the authors need to provide conclusions, meaning and interpretation so the review doesn’t read like an outline. The last is particularly valid for pg7. Line 66-67, it is unclear why CD41 is used after conversely and how the reader should connected to P2Y12. The review is hard to read and unless you are familiar with platelets and coag the more general reader would be completely lost. The review needs to be broadened and should not target s limited audience. The review would benefit from a better discussion on historical background not just listing two years. Not clear as to why the sentence on line 91-93 is included. “intuitive” should be eliminated from line 89; it is not appropriate for scientific text. Lines 164-165, what is it meant by MA/TEG is “positively correlated” with whole blood lumiaggregation. Do the authors mean comparable? Section 5 is really hard to read and understand

Author Response

The manuscript by Ranucci & Baryshnikova aims to provide a review on sensitivity of viscoelastic tests to platelet function.  The work could potentially be interesting overview but at the current stage has major issues that need to be addressed.

English language and grammar need to be revisited. Certain sentences are lacking propositions and are using colloquial words that should be omitted in scientific publication.

Reply: we have now revisited the text.

The objective of this review is to talk about viscoelastic test and platelet function. However platelet function is discussed in great limitations. In fact there is no mention of molecular targets of aspirin in platelets and how that may affect function. Overall a brief discussion of coag and platelets should be introduced.

Reply: this is an invited article for a special issue of the JCM dedicated to “Platelet Counting, Morphology Assessment and Functional Studies: Pitfalls, Uncertainties, Good Practice for Clinical Usefulness”. There are other articles in this issue that will be dedicated to the general aspects of platelet function and the pharmacological aspects of platelet inhibitors. This article was requested to address VET and platelet function, and the reader may find in the other contributions what the Reviewer is requesting

Each parameter of the various tests needs to be defined and discussed in terms of platelet function.

Reply: the VET parameters related to platelet function are widely explained. MA and MCF are explained in paragraph 1; the “platelet component” of MA and MCF is explained in paragraph 4 (now paragraph 5 in the revised version); TEG-platelet mapping is explained in paragraph 5 (now paragraph 6 in the new version). We cannot understand how we should better explain this.

Summary of findings should be not only stated but the authors need to provide conclusions, meaning and interpretation so the review doesn’t read like an outline. The last is particularly valid for pg7.

Reply: we have now expanded the conclusions (now paragraph 8) to summarize the existing evidence.

Line 66-67, it is unclear why CD41 is used after conversely and how the reader should connected to P2Y12.

Reply: we have re-written the sentence to clarify its meaning

The review is hard to read and unless you are familiar with platelets and coag the more general reader would be completely lost. The review needs to be broadened and should not target s limited audience.

Reply: as already stated, the reader will find in the same special issue of JCM all the needed background.

The review would benefit from a better discussion on historical background not just listing two years.

Reply: the historical background contains references from 1961 through 2007. The old studies are quoted to stress that in the beginning the clinicians thought that measures of clot firmness were reflecting platelet function even under the effects of antiplatelet drugs, a concept that was contradicted in the nineties. The reasons for this are explained in paragraph 3.

Not clear as to why the sentence on line 91-93 is included. “intuitive” should be eliminated from line 89; it is not appropriate for scientific text.

Reply: we have changed the sentence and eliminated the word “intuitive”

Lines 164-165, what is it meant by MA/TEG is “positively correlated” with whole blood lumiaggregation. Do the authors mean comparable?

Reply: no, we meant that there is a direct correlation. We have now changed the phrase.

Section 5 is really hard to read and understand

Reply: we have extensively revised the text.